# Influence of Na_2_O Content and Ms (SiO_2_/Na_2_O) of Alkaline Activator on Workability and Setting of Alkali-Activated Slag Paste

**DOI:** 10.3390/ma12132072

**Published:** 2019-06-27

**Authors:** Sung Choi, Kwang-Myong Lee

**Affiliations:** Department of Civil, Architectural, and Environmental Systems Engineering, Sungkyunkwan University, Jangan-Gu, Suwon 16419, Korea

**Keywords:** alkali-activator, GGBFS, Na_2_O content, Ms (SiO_2_/Na_2_O), workability, setting time

## Abstract

The performance of alkali-activated slag (AAS) paste using activators of strong alkali components is affected by the type, composition, and dosage of the alkaline activators. Promoting the reaction of ground granulated blast furnace slag (GGBFS) by alkaline activators can produce high-strength AAS concrete, but the workability might be drastically reduced. This study is aimed to experimentally investigate the heat release, workability, and setting time of AAS pastes and the compressive strength of AAS mortars considering the Na_2_O content and the ratio of Na_2_O to SiO_2_ (Ms) of binary alkaline activators blended with sodium hydroxide and sodium silicate. The test results indicated that the AAS mortars exhibited a high strength of 25 MPa at 24 h, even at ambient temperature, even though the pastes with an Na_2_O content of ≥6% and an Ms of ≥1.0 exhibited an abrupt decrease in flowability and rapid setting.

## 1. Introduction

Environmental imperatives such as the reduction of CO_2_ and conservation of natural resources are becoming issues worldwide. However, because the cement industry emits large amounts of CO_2_ and consumes much energy in the supply of raw materials and production processes, efforts are needed to reduce CO_2_ emission and energy consumption by replacing cement with supplementary cementitious materials (SCMs) such as ground granulated blast furnace slag (GGBFS) and fly ash (FA). Thus, in order to totally replace cement with GGBFS and FA attempts have been made to create zero-cement concrete by the alkali activation of GGBFS and FA [1,2,3,4,5].

Alkali-activated slag (AAS) concrete, which is produced by using an alkali activator on GGBFS, is a typical zero cement concrete. As GGBFS is calcium aluminosilicate vitreous, it has strong latent hydraulic properties. However, alkaline (pH = 12) activators are needed to stimulate GGBFS. After the GGBFS is stimulated, Si^4+^, Al^3+^, Ca^2+^, Mg^2+^, and Na^+^ ions are eluted from GGBFS and the reaction of GGBFS proceeds. It should be noted that GGBFS activated in this way has very fast reaction rates due to chemical ionic reactions [6]. Alkaline activators largely impact the properties of AAS concrete and thus, their characteristics such as the alkali concentration, the dosage, and the water-to-solid ratio should be examined prior to the mixture design of AAS concrete [7,8]. In general, soluble alkali or alkali salts can be used as alkaline activators. When considering the strength development and economic efficiency of AAS concrete, the most commonly used activators are Na_2_SiO_3_ (sodium silicate) and NaOH (sodium hydroxide). A better understanding of the effects of such alkaline activators on the reaction mechanisms of alkali-activated slag could indicate ways to optimize the use of alkaline activators.

Recently, research has been conducted to improve the performance of AAS concrete by blending two types of alkaline activators [9]. When such a binary alkaline activator is used, the characteristics possessed by each alkaline activator can be complementarily utilized [10,11]. NaOH promoted initial reaction of AAS, and Na_2_SiO_3_ was effective in increasing the strength of AAS concrete. Chang [12] evaluated the setting characteristics of silica-activated pastes depending on the type of alkaline activator used and concluded that setting time significantly reduced as the amount of alkaline activator increased. Zuo et al. [13] showed that though the reaction degree increased as the amount of alkaline activator increased, the reaction degree no longer increased when the amount of alkaline activator was greater than 6 wt.% of cement. Krizan and Zivanovic [14] reported that high strength could be expected when the Ms (SiO_2_/Na_2_O) were in the range of 0.6–1.5. As described above, as the amount of alkaline activator increases, the reaction of GGBFS increases and the strength of AAS concrete also increases. Therefore, it is necessary to determine an appropriate range for the usage amount of the activator for the mix design of AAS concrete. In particular, the effects of Na_2_O and SiO_2_ concentrations in alkaline activators on the strength development of AAS concrete should be examined in order to effectively use the alkaline activators.

AAS concrete, which stimulates GGBFS using alkaline activators, rapidly increases in strength in the early ages and exhibits high strength of 40 MPa or more at 28 days [15,16]. However, because the initial reaction rate of GGBFS by alkaline activators is extremely high, several problems with AAS, such as the flowability being lost initially or rapid setting occurring, must be improved for it to be used in a practical setting [17,18,19]. To address this, the effects of the alkaline activator on the fresh properties of AAS concrete such as workability and setting should be examined, even though the AAS activation mechanism at early ages has not been clearly identified.

In this study, in order to identify the characteristics of AAS pastes, a binary alkaline activator that was a blend of NaOH and Na_2_SiO_3_ was used. Nine types of AAS paste mixture, varying by their Na_2_O content and the ratio of Na_2_O to SiO_2_ (Ms), were tested. First, the reactivity of the AAS was evaluated for 24 h through the calorimetric measurement of the AAS pastes, and the workability and setting time were measured. Furthermore, to quantitatively analyze the instant at which flowability is lost and setting occurs in the AAS pastes, the viscosity and ultrasonic pulse velocity of the pastes were measured. Lastly, the compressive strength of the AAS mortars was measured up to 24 h. Consequently, it was found that the workability and setting of AAS pastes and the initial strength development of the AAS mortars were significantly affected by Na_2_O content and Ms.

## 2. Materials and Method

### 2.1. Materials and Alkaline Activator

GGBFS obtained from POSCO (Pohang, Korea) was used as a binder. The chemical composition of the GGBFS determined with X-ray fluorescence (XRF, BRUKER S8 Tiger, Billerica, MA, USA) is given in Table 1. The GGBFS was composed of 44.0% CaO, 33.7% SiO_2_, 13.8% Al_2_O_3_, and 5.2% MgO and thus, its basicity coefficient (Kb = (CaO + MgO)/(SiO_2_ + Al_2_O_3_)) was 1.04, which is similar to the neutral value of 1.0 for the ideal alkali activation [12]. The hydration modulus (HM = (CaO + MgO + Al_2_O_3_)/SiO_2_) was 1.87, which was 33.5% higher than the required value of 1.4 for sound hydration properties of GGBFS [12,19]. Table 2 shows the physical properties of GGBFS. The density and Blaine air permeability of the GGBFS were 2.90 g/cm^3^ and 4253 cm^2^/g, respectively. Figure 1 shows the particle size distribution of GGBFS measured with a laser particle size analyzer (PSA, BECKMAN COULTER LS 13 320, Brea, CA, USA). The medium particle size (d_50_), which represents the particle size of a cluster of particles, was 10.2 μm, and d_10_ and d_90_ were 1.4 μm and 33.2 μm, respectively. The dissolution of GGBFS is dominated by small particles. That is, particles >20 µm react slowly, while particles <2 µm react completely after 24 h in alkali-activated binders [20,21].

The physical properties of the fine aggregates that were used and the mortar production were measured complying with ASTM C 128-15 [22] and ASTM C 33-13 [23]. The fine aggregates were the crushed sands of granite, and their density and absorption rate were 2.62 g/cm^3^ and 1.04%, respectively. Figure 2 shows the particle size distribution of fine aggregates. The max. size of the fine aggregate was 4.76 mm, and the fineness modulus was 2.74.

Two types of alkaline activator, sodium hydroxide (98% purity, Duksan, Ansan, Korea) and sodium silicate (Type 3 industrial water glass, Na_2_SiO_3_, Ganachem, Ulsan, Korea) were used to activate the GGBFS. The chemical composition of sodium silicate solution was SiO_2_ = 28.3 wt.%, Na_2_O = 9.3 wt.%, and H_2_O = 62.4 wt.%. A 2 M sodium hydroxide solution was prepared by mixing with deionized water. Furthermore, because NaOH would release heat when mixed with water, in order to cool down to ambient temperature, the alkaline activator was stored in a chamber at 20 °C and the relative humidity (RH) of 60% for 24 h prior to casting AAS pastes. The pH of the activators was ranged 13.2–13.6, as measured using a pH meter (HORIBA, LAQUA 74bW, Kyoto, Japan).

### 2.2. Mixture Design

Table 3 shows the mixture design of the AAS pastes used to evaluate the workability and setting of the AAS pastes. In this study, a liquid-to-binder ratio (l/b, binder = GGBFS content) was fixed as 0.42 and the Na_2_O content and Ms of alkaline activator were varied. It should be noted that the water content of Na_2_SiO_3_ was included in the required water content. Also, NaOH was added as an alkali activator in order to adjust Ms according to the dosage of Na_2_SiO_3_. The Na_2_O content was set to 4%, 6%, and 8% according to the weight of the GGBFS, and the three levels of Ms were considered at 0.75, 1.00, and 1.25. Thus, nine AAS paste mixtures were designed, and the amount of alkaline activators for each mixture was determined, as shown as Table 3. In the mixture section in the table, the first and second components represent Na_2_O content and Ms, respectively. To produce the AAS mortar mixture, twice as much fine aggregate as GGBFS was added to the AAS paste mixture. 

### 2.3. Test Method

#### 2.3.1. Calorimetric Measurement

GGBFS generates heat during reacting with alkaline activator. To measure the heat, GGBFS and water were stored for 24 h in the calorimeter equipment at 20 °C. The AAS paste mixtures were prepared by mixing the GGBFS, water, and alkaline activators using an internal propeller for 5 min. The mixture was placed into the calorimeter (Tokyo-Riko, Three-point multi-purpose conduction calorimeter, Tokyo, Japan). The temperature range of the equipment was 5–60 °C, and the temperature stability was 5 × 10^−3^ °C/5 °C. The amount of heat release was measured immediately after mixing the AAS paste up to 24 h.

#### 2.3.2. Workability

The workability of AAS paste mixtures was tested using the mini slump test method. Figure 3 shows a mini slump setup and flow measurement of AAS paste. The mold for the mini slump test was made of brass, and its height was 50 mm, the top opening was 70 mm, and the bottom opening was 100 mm. After filling the mold with the paste mixture, the mold was lifted and the spread length of the mixture was measured. The mini slump of the AAS paste was tested a total 13 times for each paste, from 0 min through 60 min at 5 min intervals after casting the paste complying with ASTM C 1437-15 [24]. In order to eliminate the possibility of measurement errors in the test due to moisture evaporation from the surface of the paste, the paste was remixed for 30 s before mini slump test.

#### 2.3.3. Setting Time

The Vicat test (per ASTM C 191-18a [25]) test was carried out to measure the setting time of the AAS pastes inside a room chamber at 23 °C and 50% (RH). The Vicat initial time of setting is defined as the elapsed time between the initial contact of cement and water and the penetration of 25 mm. The Vicat final time of setting end point is defined as the first penetration measurement that did not mark the specimen surface with a complete circular impression.

#### 2.3.4. Rheology and Ultrasonic Pulse Velocity (UPV)

The rheology and UPV tests were adopted to evaluate the timing at which the AAS paste lost its flowability, by measuring the viscosity and ultrasonic pulse velocity of the AAS paste according to the elapsed time. The rheology test measured viscosity through a paste viscometer using the friction resistance between the spindle and the paste rotating at a speed of 5 rpm. To determine the moment that flowability was lost in the AAS paste, the time to reach maximum measurable viscosity was recorded. The UPV test was carried out by placing the oscillator and receiver on acrylic boxes (thickness: 10 mm; ultrasonic velocity of acryl: 2740 m/s) placed 20 mm apart, filling the container with paste, and measuring the transmission time of ultrasonic waves through the oscillator and receiver at intervals of 30 s with the measurement frequency of 5 kHz.

#### 2.3.5. Compressive Strength

To investigate the influence of Na_2_O content and Ms on the compressive strength development of AAS mortars up to 24 h, uniaxial compression test was conducted. A mortar mixer was used for mixing 2-liter batches. After premixing GGBFS and fine aggregates for 30 s, the alkaline activator and water were added at low speed. Mixing continued at low speed for 90 s and at high speed for 1 min. Cube mortar specimens (50 × 50 × 50 mm^3^) were prepared complying with ASTM C109-16a [26] and cured in a chamber at 20 ± 2 °C and the relative humidity (RH) of 60 ± 2%, for curing at the environmental condition similar to general ambient condition. Compressive strength of the mortar specimens was measured at 12 h, 18 h, and 24 h after casting the AAS paste. The compressive strength was obtained as an average value of three AAS mortar specimens.

## 3. Results and Discussion

### 3.1. Heat Release of Pastes

Figure 4 shows the calorimetric curves for the rate of heat release according to Na_2_O content. The AAS pastes, excluding the paste with 4% Na_2_O, had two peaks of heat [13]. When Na_2_O content was greater than 6% of binder weight, the first peak of heat occurred at 30–60 min, which resulted mainly from the wetting and partial dissolution of GGBFS by NaOH and partly from the reaction of GGBFS by Na_2_SiO_3_. The second peak was observed between 6 h and 12 h, which corresponds mainly to the formation of reaction products [27]. That is, when Na_2_O content was fixed to be 6% or 8%, the first peak of heat increased as Ms increased, in other words, SiO_2_ content increased. In particular, the paste with 6% Na_2_O content showed a relatively high first peak and a clear secondary peak. Ravikumar and Neithalath [28] reported a similar result, stating that at constant Na_2_O content, the Ms was a main factor in increasing both the speed and the magnitude of the first peak of heat. For the pastes with alkaline activators of 4% Na_2_O content, the first peak of heat was relatively low and the secondary peak occurred after the induction since the amount of alkaline activators was less. This is a similar tendency to the rate of heat release curve when only Na_2_SiO_3_ was used [27].

### 3.2. Workability

Figure 5 shows the mini slump test results for AAS pastes with a l/b of 0.42. The paste without an alkaline activator demonstrated an initial flow of 182 mm, while the pastes with alkaline activators showed an initial flow of 205–350 mm. All of the AAS pastes with alkaline activators of 6% and 8% Na_2_O content lost their flowability within 15 min. The flowability decreased more rapidly as Ms increased; specifically, for the 8-1.25 paste, with the highest Na_2_O content and Ms, hardening occurred just after 5 min after mixing. As mentioned in Section 3.1, this is because the activation and reaction of GGBFS by the alkali activators occurred rapidly at the early ages. Once the AAS paste lost its flowability, the pastes began to harden quickly. These tendencies appeared as Ms increased because more reaction products of the GGBFS were produced by the sufficient supply of SiO_2_ [14].

The AAS paste with alkaline activators of 4% Na_2_O content maintained a flow of >200 mm for 1 h after mixing. Similarly, Puertas et al. [29] reported that the analysis of the flow performance of AAS pastes with alkaline activators of less than 5% Na_2_O content showed that when Ms was lower than 0.8, the flowability of the paste was maintained for 60 min.

The mini slump test results of the AAS pastes with l/b of 0.45 or 0.48 are shown in Figure 6. The AAS pastes with l/b of 0.45 and 0.48 lost flowability at 20 min and 25 min, respectively, while the AAS pastes with l/b of 0.42 lost flowability within 15 min. When l/b increased from 0.42 to 0.48, flowability was enhanced by just 10 min.

It was also observed that the higher the Ms, the lower the initial flow of the AAS paste. This is because as the Ms increased, workability was reduced due to the increase of the reaction products of the GGBFS [14,30,31]. However, the times at which the AAS pastes lost their workability were nearly identical regardless of the Ms.

### 3.3. Setting Time

Table 4 shows the Vicat test results of the AAS pastes on the initial and final setting time. The setting of the AAS paste was accelerated by the increase of activation and reaction of the GGBFS when the dosage of alkaline increased. Similarly, Chang [12] reported that the setting time of the AAS pastes correlated with the sum of the Na_2_O (from NaOH and Na_2_SiO_3_) and SiO_2_ (from Na_2_SiO_3_) concentrations of the alkali activator. For AAS pastes with less than 8 g of alkaline (per 100 g of GGBFS), setting time significantly increased as the dosage of alkaline was low. For AAS pastes with more than 10 g of dosage of alkaline, setting was accelerated; initial setting occurred within 1 h, as shown in Figure 7. When the dosage of alkaline increased to more than 14 g, final setting occurred within 30 min.

### 3.4. Viscosity Measurement Limit and Ultrasonic Pulse Velocity

Figure 8 shows the rheology test results of the AAS pastes with Na_2_O content and Ms. The viscosity measurement limit (VML) refers to the point at which flowability is lost in the fluid state. As described in Section 3.2, the flow of the paste showed a tendency for the flowability to decrease abruptly immediately before the loss of flowability, and the shear stress rapidly increased 3 to 5 min before reaching VML. As shown in Table 4, the min. and max. VML times of the AAS paste with ≥6% Na_2_O content were 540 s for 8-1.25 and 1040 s for 6-0.75, respectively. These VMLs were much correlated with the time at which AAS pastes lost their flowability, as shown in Figure 5b,c, since the GGBFS reaction was quickened by tens of minutes after the mixing. 

The AAS paste with 4% Na_2_O retained its flowability for up to 1 h in the mini slump test, but its VML was only 15–20 min. This was because the AAS paste was remixed for 30 s before the measurement of the flow, while in the rheology test, the shear stress of the AAS paste was measured continuously for 30 min without remixing.

Figure 9 shows the ultrasonic pulse velocity (UVP) measurements of the AAS pastes; the initial ultrasonic velocity of the paste was 302.5–405.0 m/s regardless of the mixture type [32,33]. This indicates that the paste was in a fluid state. Afterwards, as the paste lost its flowability, the UPV increased sharply, and during certain periods of time before VML was reached, the UPV increased intermittently rather than continuously. Table 5 summarizes the UPVs of the time at which the VML was reached. The UPVs of the AAS pastes increased gradually during the reaction of the GGBFS [34]. The rheology test results showed that when the AAS paste reached the VML, the UPVs were confirmed to be 900–1000 m/s. After the VML, the UPVs increased steadily, and then, the increase rate of UPVs differed in accordance with the Na_2_O content and Ms. It is observed in Figure 9 that after the VML, UPVs of the AAS pastes with an Na_2_O content of ≥6% and Ms of ≥1.00 rapidly increased, but UPVs of the AAS pastes with 4% Na_2_O content or Ms of 0.75 modestly increased.

### 3.5. Compressive Strength

Figure 10 shows the compressive strength development of the AAS mortars. The AAS mortars with a 4% Na_2_O content had a compressive strength of <1 MPa up to 24 h regardless of Ms, implying that the increase of SiO_2_ content did not significantly affect the compressive strength development. This is consistent with the very low heat release for the initial 24 h, as shown in Figure 4a. On the other hand, although the mortar with ≥6% Na_2_O showed an increase in strength as time elapsed, the compressive strength differed in accordance with the Ms. When Ms remained constant, the compressive strength development of the AAS mortars with 8% Na_2_O was very similar to that of the AAS mortars with a 6% Na_2_O. This was in accordance with the results of Zuo et al. [13], who reported that when Na_2_O was greater than 6%, the reaction degree did not increase significantly with time, which in turn kept compressive strength relatively constant.

## 4. Conclusions

The objective of this study was to investigate the effects of Na_2_O content and Ms of the alkali activator on the workability and setting of AAS pastes. To evaluate the fresh properties of the pastes, mini slump, Vicat test, and rheology/UPV test were applied. Based on the experimental results, the following conclusions can be drawn:The Na_2_O content and Ms strongly affected the rate of heat release, workability, and setting time of alkali-activated pastes. In particular, when alkaline activators with Na_2_O content of ≥6% (Ms = 0.75–1.25) were used, the activation and reaction of the GGBFS were enhanced because of the effects of high dosage of alkaline (Na_2_O + SiO_2_), which resulted in workability rapidly decreasing and setting quickly occurring. However, the Na_2_O content and Ms did not significantly affect the compressive strength development of the AAS mortars when the Na_2_O content increased from 6% to 8%. On the other hand, for the pastes with Na_2_O content = 4%, the rate of heat release was quite low for the initial 24 h and setting was delayed up to 4 h and 40 min, and even the workability was maintained for more than 1 h.In AAS pastes with high Na_2_O content, setting occurred within a few minutes after flowability was lost, and thus, it was not easy to measure the flowability with the existing mini slump test. However, since the changes in viscosity of the paste can be continuously monitored over time using the rheology/UPV test, the measurement of VML and UPV could allow the quantitative assessment of the time at which flowability was lost and consequently, to predict the initial setting time of the paste.For the production of AAS concrete with high early strength of 25 MPa at 24 h, it is suggested from the test results that the Na_2_O content and Ms of the alkaline activator should be as much as 6% and 1.0, respectively. However, to secure the workability of AAS paste, chemical admixtures such as superplasticizers should be used to adequately control the workability and setting in highly alkaline environments.

## Figures and Tables

**Figure 1 materials-12-02072-f001:**
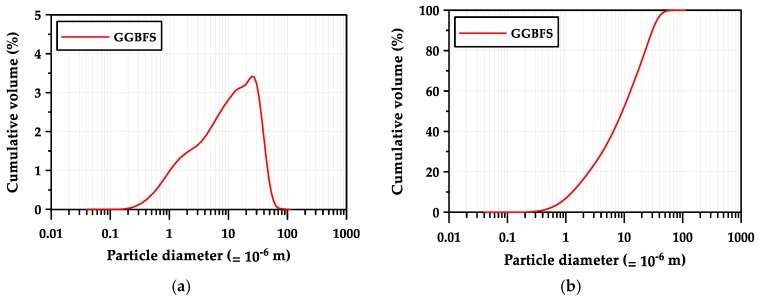
Particle size characteristics of GGBFS: (**a**) particle size distribution, (**b**) cumulative particle size distribution.

**Figure 2 materials-12-02072-f002:**
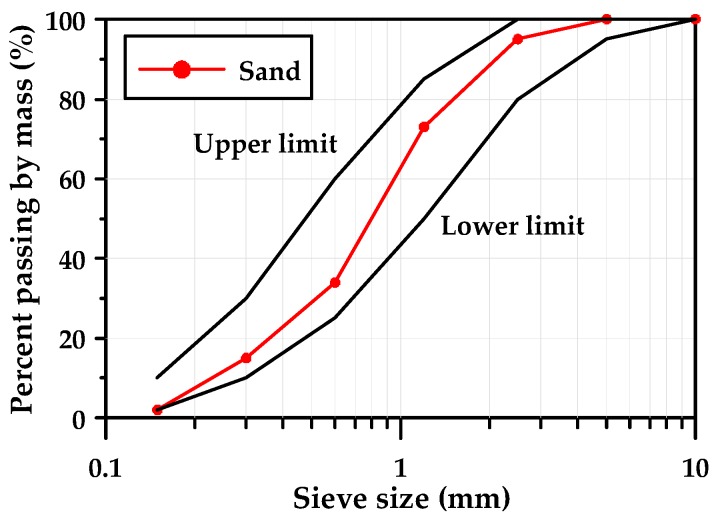
Particle size distribution curve of fine aggregates.

**Figure 3 materials-12-02072-f003:**
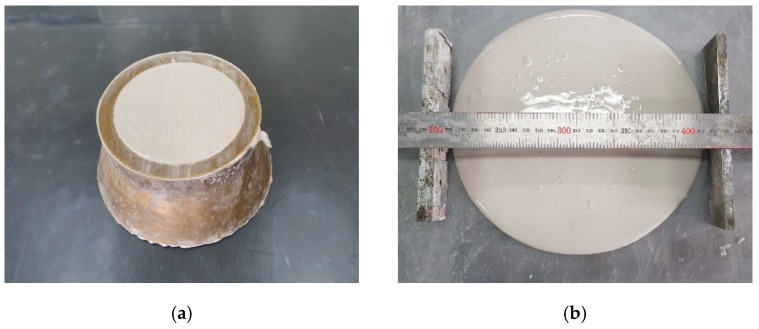
A mini slump test: (**a**) test setup, (**b**) flow measurement of AAS paste.

**Figure 4 materials-12-02072-f004:**
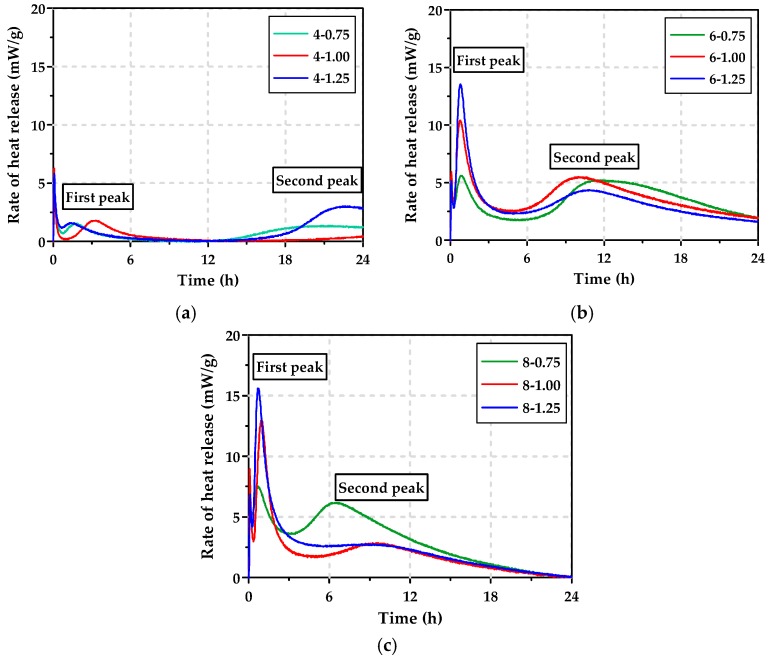
Rate of heat release curves of AAS pastes for 24 h: (**a**) Na_2_O content = 4%, (**b**) Na_2_O content = 6%, (**c**) Na_2_O content = 8%.

**Figure 5 materials-12-02072-f005:**
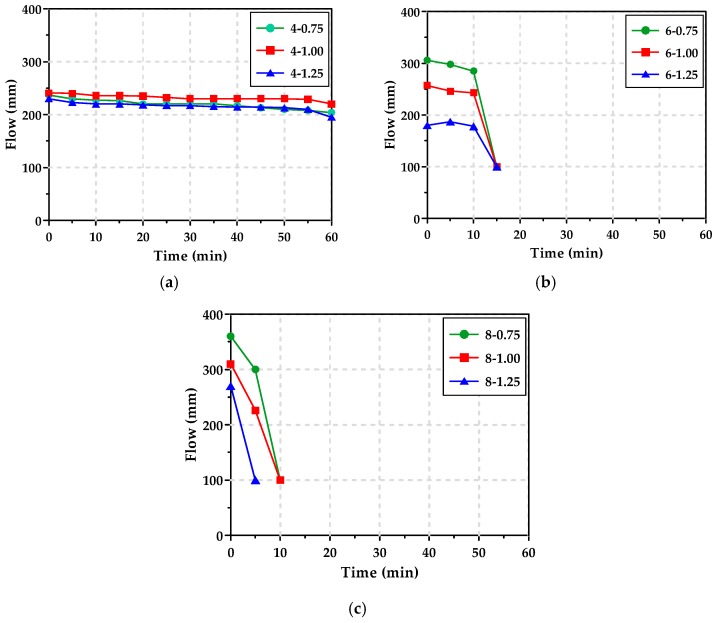
Mini slump curves of AAS pastes: (**a**) Na_2_O content = 4%, (**b**) Na_2_O content = 6%, (**c**) Na_2_O content = 8%.

**Figure 6 materials-12-02072-f006:**
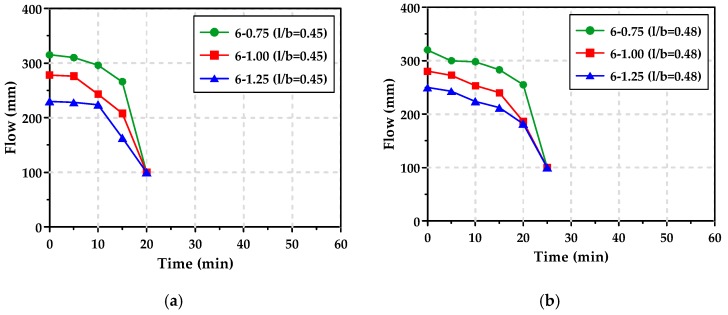
Mini slump curves of AAS pastes: (**a**) l/b = 0.45, (**b**) l/b = 0.48.

**Figure 7 materials-12-02072-f007:**
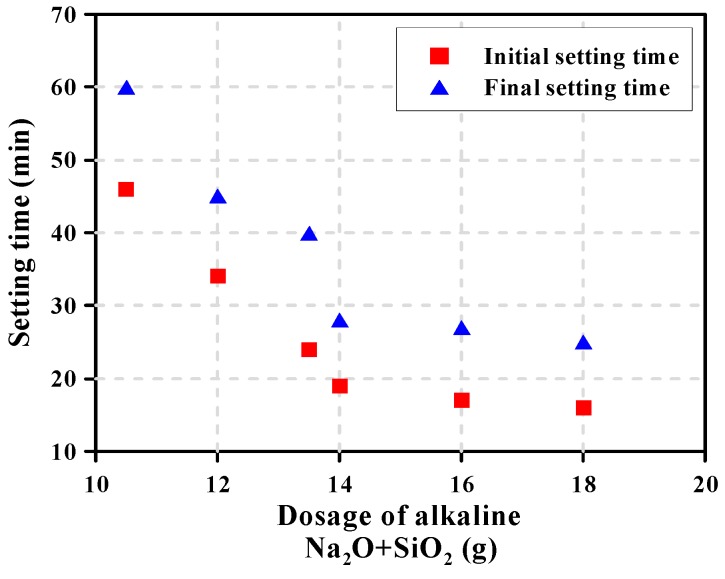
Setting time of AAS paste by the dosage of alkaline.

**Figure 8 materials-12-02072-f008:**
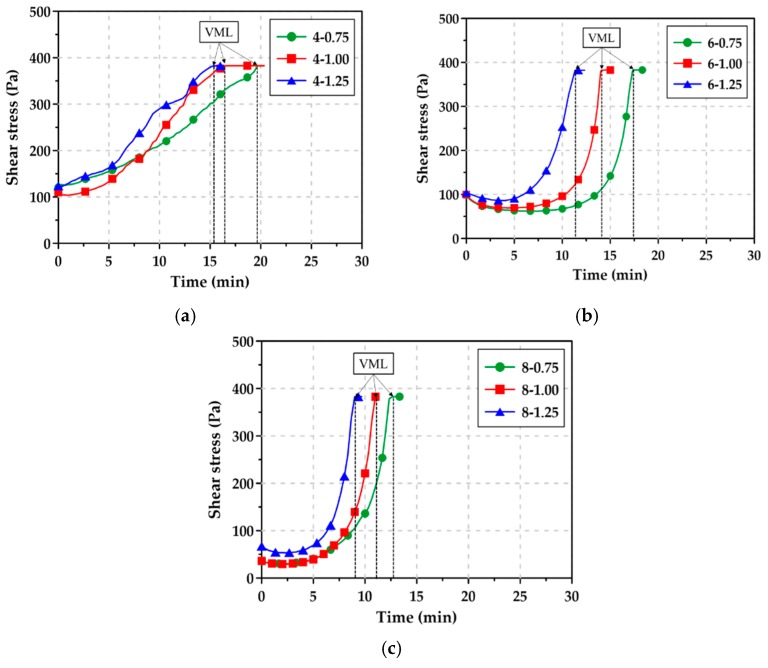
Rheology test curves for AAS pastes: (**a**) Na_2_O content = 4%, (**b**) Na_2_O content = 6%, (**c**) Na_2_O content = 8%.

**Figure 9 materials-12-02072-f009:**
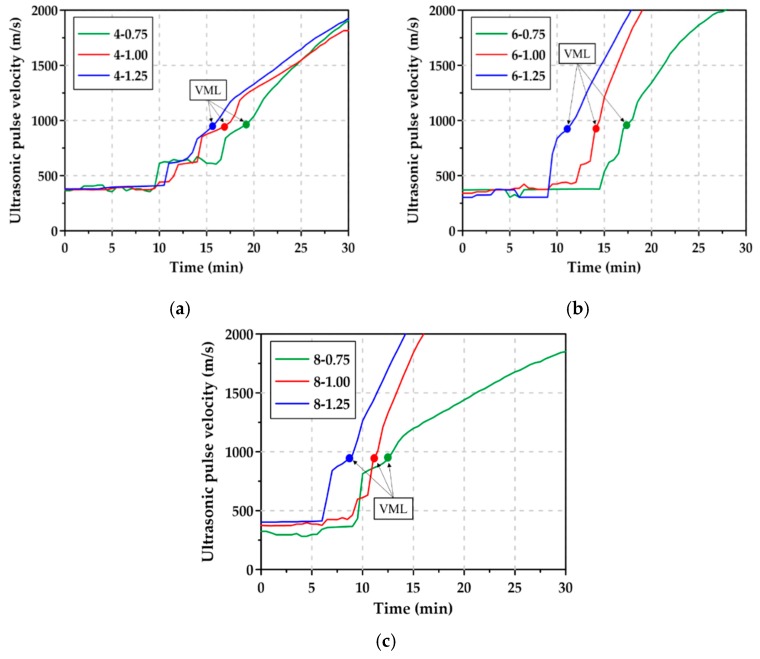
UPV curves for AAS pastes: (**a**) Na_2_O content = 4%, (**b**) Na_2_O content = 6%, (**c**) Na_2_O content = 8%.

**Figure 10 materials-12-02072-f010:**
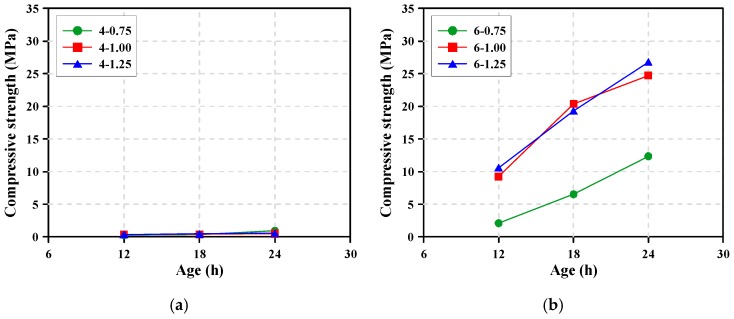
Compressive strength development of AAS mortar specimens cured in a chamber at 20 °C: (**a**) Na_2_O content = 4%, (**b**) Na_2_O content = 6%, (**c**) Na_2_O content = 8%.

**Table 1 materials-12-02072-t001:** Chemical composition of ground granulated blast furnace slag (GGBFS) (wt.%).

Type	CaO	SiO_2_	Al_2_O_3_	MgO	SO_3_	TiO_2_	K_2_O	M_n_O	Fe_2_O_3_
GGBFS	44.0	33.7	13.8	5.2	1.2	0.7	0.5	0.2	0.1

**Table 2 materials-12-02072-t002:** Physical properties of GGBFS.

Type	Density (g/cm^3^)	Blaine (cm^2^/g)	Particle Size Distribution (μm)			
			Mean	d_10_	d_50_	d_90_
GGBFS	2.90	4253	14.2	1.4	10.2	33.2

**Table 3 materials-12-02072-t003:** Mixture design of alkali-activated slag (AAS) pastes with respect to 100 g of binder.

Mixture	Na_2_O	SiO_2_	Alkaline (Na_2_O + SiO_2_)	Ms (SiO_2_/Na_2_O)	Alkali Activator		l/b
					NaOH	Na_2_SiO_3_	
4-0.75	4.0	3.0	7.0	0.75	3.89	10.60	0.42
4-1.00	4.0	4.0	8.0	1.00	3.47	14.13	
4-1.25	4.0	5.0	9.0	1.25	3.04	17.67	
6-0.75	6.0	4.5	10.5	0.75	5.83	15.90	
6-1.00	6.0	6.0	12.0	1.00	5.20	21.20	
6-1.25	6.0	7.5	13.5	1.25	4.56	26.50	
8-0.75	8.0	6.0	14.0	0.75	7.78	21.20	
8-1.00	8.0	8.0	16.0	1.00	6.93	28.27	
8-1.25	8.0	10.0	18.0	1.25	6.08	35.34	

**Table 4 materials-12-02072-t004:** Vicat test results of AAS pastes.

Mixture	Initial Setting Time (min)	Final Setting Time (min)
4-0.75	280	355
4-1.00	65	112
4-1.25	33	53
6-0.75	46	60
6-1.00	34	45
6-1.25	24	40
8-0.75	19	28
8-1.00	17	27
8-1.25	16	25

**Table 5 materials-12-02072-t005:** Ultrasonic pulse velocities (UPVs) and viscosity measurement limits (VMLs) of AAS pastes.

Mixture	VML (min:s)	UPV (m/s) at VML
4-0.75	19:40	990.75
4-1.00	16:20	929.75
4-1.25	15:20	939.00
6-0.75	17:20	967.75
6-1.00	14:00	909.50
6-1.25	11:20	911.75
8-0.75	12:20	936.50
8-1.00	11:00	909.50
8-1.25	9:00	972.50

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
