# Peer review of "Influence of Na2O Content and Ms (SiO2/Na2O) of Alkaline Activator on Workability and Setting of Alkali-Activated Slag Paste"

_materials, 2019, doi:10.3390/ma12132072_

Round 1

Reviewer 1 Report

Title: Influence of Na2O Content and Ms (SiO2/Na2O) of Alkali Activator on Fluidity and Setting of Alkali-activated Slag Paste

The paper investigates the influence of Na2O content and Ms (SiO2/Na2O) of alkali activator on fluidity and setting of alkali-activated slag paste. A topic is of interest for the readers of Materials. However, interpretation of results is limited and there are many questions with regard to test methods and results which suggest major revision of this paper.

The main impressions are:

the introduction (background and objectives of the research) needs rewriting by clearly stating the reasons for this study. References’ list is very poor. More references are needed to be included, because there is a plenty of studies which studied similar behavior of alkali activated pastes such as in this paper. I suggest to authors to find them and add them to their work and reinforce better their introduction.

missing information about units, descriptions of mixing, particle size distribution of GGBFS and fine sand, in Materials and Methods section. Please, add them.

in-depth analysis and discussions of the results. Results are mainly reported without explanations for certain behavior

not sufficient experimental details and discussions. In all your Results section you do not compare your results with results obtained from literature. Please, compare your results with other researches. What did you find different, compared to available literature? Why your results are important? What they show compared to other people’s papers?

I would be glad to revise the improved version of this paper, since it has a large potential to contribute to understanding of the influence of Na2O content and Ms (SiO2/Na2O) of alkali activator on fluidity and setting of alkali-activated slag paste. It can also have large impact on practical applications. I also suggest to add more discussions, correlate results better, make clear pictures.

Abstract:

Introduction

Introduction has to be enriched with other references related to the performed research of authors such as:

Haha, M.B.; Le Saout, G.; Winnefeld, F.; Lothenbach, B. Influence of activator type on hydration kinetics, hydrate assemblage and microstructural development of alkali activated blast-furnace slags. Cem. Concr. Res. 2011, 41, 301–310.

Materials and methods

By calculating the basicity coefficient which is calculated using wt% of oxides (Kb=(CaO+MgO)/(SiO2+Al2O3)), I get 1.04, and you reported the value of 1.7? Why 1.7? Please, check it!

How did you calculate water absorption of fine aggregates?

What was the max size of your fine aggregates? Was it 4 mm? Please, write this information in your section Materials and methods.

Line 62 Please write: alkaline activators, and not alkali activators.

Line 69 From where you derived this target flow of 180±5 mm, please insert the reference for this target flow? Why do you want to be 180±5 mm? Then where you can apply this paste with this target flow?

Table 2 Mix proportions of AAS pastes, in which unit you give water, GGBFS, NaOH, Na2SiO3, are these numbers in wt%, please add this to the table. The same you should report for Table 1, are the oxides given in wt%?

Line 74 Explain to readers, what is Ms?

Lines 75 and 76 Please, explain how did you mix and make mortars, which sample size was cast, which type of curing, how long?

Fine aggregate, was it quartz? Was it standard fine aggregate, what was the particle size distribution?

GGBFS, what is the particle size distribution, please add, such as . What is mean particle size of GGBFS? What is the D50 of GGBFS?

Particle size distributions of GBFS and fine sand is needed to be added to this paper, such as you can see in paper;

Nedeljković, Marija, Zhenming Li, and Guang Ye. "Setting, strength, and autogenous shrinkage of alkali-activated fly ash and slag pastes: Effect of slag content." Materials 11, no. 11 (2018): 2121.

2.2.1

You need to improve description of the test method “Heat of hydration test”

How was the heat evolution investigated? Which standard was used?

2.2.2

Do you have a photo of the set up for mini-flow test? It would be good to add it to the paper to illustrate how did you perform fluidity test? Why do you call it actually “fluidity test”? This is just a workability test what you performed in your work.

2.2.3

What was the temperature and  relative humidity of your room where you performed your setting time measurements?

2.2.4

Why did you use RH of 60±2% for curing of your cubes for compressive strength measurements? Please, explain it in section 2.2.4

3.1

Do you plot heat of hydration on Y-axe ? Then it should be that the unit is mW/g, and not J/g.

Why did you measure the heat of hydration only for an hour? Do you have measurements data for longer periods?

Lines 134-137 How do you prove this hypothesis? Do you have reference for all these sentences and assumption for Si4+ ions increase in the mixing water?

Line 155 What is W/B? Please, make clear that readers know your abbreviations. What is W, what is B?

If W/B is a water-to-binder ratio then you should express not in [%] but as dimensionless, 0.42, …

Lines 162 and 163 Again, how do you prove these hypotheses? How do you know there is an increase of amount of reaction products?

Please, do not write hydration products, it is more correct to write “reaction products of AAS”.

Figure 4 is missing more descriptions, please add them, otherwise Figure 4 may not be needed in the manuscript.

Difference between setting times of paste with 4-0.75 and 4-1.25 is quite significant? You can explain this?

Conclusions

You need to rewrite this part and say clearly how your research contributes to the current understanding of the researched topic. What I should learn from it and what are the scientific merits of your results?

Author Response

See attatchment.

Reviewer 2 Report

The paper concerns the investigations mainly about the fresh property state of the alkali-activated slag paste. The experiments and methods are chosen correctly to properly describe the scientific problem. The paper is written in a very simple way. It is a good technical paper, however, in order to publish it in the Materials, there must be more scientific interpretation of the results obtained – this is the main shortcoming of the paper presented. Detailed comments are listed below:

1.      The paper should be checked by native English speaker in order to improve the style; there is also few syntax and grammar errors.

2.      Introduction section - The section should be much more extensive. Performing a literature review referring only to 14 references does not provide an opportunity to objectively present the given issue.

3.      Line 59-62 - please give a methodology, how all of these parameters were tested?

4.      Compressive strength test - on how many samples has the compressive strength test been carried out? How many samples were tested for a given recipe? Without information on the distribution of the results, the reliability of the results obtained can not be assessed.

5.      The authors present the results well, but there is no detailed, scientific discussion of why this is happening; wanting to publish an article in the Materials, the authors should attempt to explain the results obtained scientifically, bearing in mind the structure of the material and changes occurring in it as a result of the binding process. There is the main weakness of the article.

Author Response

See attatchment.

Round 2

Reviewer 1 Report

Dear authors, thank you for your answers to the comments. The manuscript has been substantially improved. The revised version can be accepted for publication.

Reviewer 2 Report

All suggestions have been included. I accept the paper for publication.